# Influential Mechanism of $Cr_3C_2$ Content and Process Parameters on Crack Generation of $Fe_3Al/Cr_3C_2$ Composites

Yingkai Feng [1], Shaoquan Liu [1,*], Jiayu Zhao [1], Lin Zhao [2,*] and Zhenbo Liu [2]

1   Chinese Institute of Coal Science, Beijing 100013, China; fengyingkai1029@126.com (Y.F.); 15650700056@163.com (J.Z.)
2   Welding Institute, Central Iron and Steel Research Institute, Beijing 100081, China; lzbzzhy@163.com
*   Correspondence: 13020003390@163.com (S.L.); hhnds@aliyun.com (L.Z.)

**Abstract:** In this paper, a brake cylinder coating comprising composite material of $Fe_3Al$ and $Cr_3C_2$ mixed powder was prepared by laser cladding onto carbon structural steel. The microstructure of the cladding materials was investigated through X-ray diffraction (XRD), scanning electron microscopy (SEM), energy dispersive spectroscopy (EDS), and transmission electron microscopy (TEM). The influence mechanism of the $Cr_3C_2$ content and process parameters on crack generation of $Fe_3Al/Cr_3C_2$ composites were studied. The crack is mainly caused by the mismatch between the plastic phase of $Fe_2AlCr$, the brittle phase of $Fe_3Al$, and $Cr_7C_3$ in the cladding layer. With the increase in $Cr_3C_2$ content, $Fe_2AlCr$ and $Cr_7C_3$ in the cladding layer increased, while $Fe_3Al$ decreased. When the content of $Cr_7C_3$ is low, the cladding layer mainly reflects the plastic increase brought by $Fe_2AlCr$; when the content of $Cr_7C_3$ is too much, the property of the cladding layer is reflected as the brittleness of $Cr_7C_3$. The number of cracks reached the lowest value at 15 wt.%. The process parameters mainly affect the dilution rate of the cladding layer and thus affect the content of $Fe_2AlCr$.

**Keywords:** carbide behavior; $Fe_3Al/Cr_3C_2$ composites; laser cladding; crack generation





## 1. Introduction

Coal machinery equipment is currently facing common operating conditions such as high temperature, corrosive environment, and excessive wear; therefore, the requirements for materials are relatively strict. $Fe_3Al$ alloy and its composite materials have good resistance against wear, high temperatures, and oxidation [1–4]. Meanwhile, due to the low cost of Fe and Al elements, they will bring significant economic benefits in large-scale industrial applications. Therefore, this material is expected to become an important new material in the future coal machinery field.

$Cr_3C_2$ has high thermal hardness, good corrosion resistance, and high oxidation resistance, which are compatible with the thermal expansion coefficient of most alloys. Therefore, the addition of $Cr_3C_2$ to the matrix as a hard-strengthening phase will greatly improve the wear resistance of a material. Wang et al. [5] prepared a series of coatings containing 15 wt.% $TiFe/Cr_3C_2$ powder in situ synthesized TiC by plasma-transferred arc cladding. It was found that a coating with a Ti/C ratio of 2:1 was the optimal ratio, with moderate transition zone and hardness, defect-free interface, and best high-temperature wear resistance. Fan et al. [6] studied the phase composition, microstructure, mechanical properties, and friction and wear of $Ni_3Al$-based coatings at 25 °C to 800 °C with different $Cr_3C_2$ concentrations (0 wt.%, 10 wt.%, 15 wt.%, and 20 wt.%) behavioral impact. The second phase strengthening of $Cr_3C_2$ and the solid solution in the $Ni_3Al$ matrix phase significantly improve the hardness and toughness of the coating, thereby enhancing its wear resistance.

Laser cladding is a method of adding cladding materials to the surface of the substrate and using high-energy density laser beams to melt them together with the thin layer on the

surface of the substrate, forming a metallurgically bonded additive cladding layer on the surface of the substrate. Due to the more concentrated energy of laser cladding, the base material's heat-affected zone is small. In particular, when used to melt different materials, the characteristics of laser cladding are vastly superior to other heat sources [7].

The research on $Fe_3Al$ alloy and $Fe_3Al$ matrix composite materials has mainly focused on $Fe–Al/Al_2O_3$, $Fe–Al/WC$, and other composite materials. In the research on $Fe–Al/Cr_3C_2$ composite materials, only thermal spraying coating technology has been largely considered. Systematic investigations on the phase composition and microstructure of $Fe_3Al/Cr_3C_2$ composite materials, as well as the type and cause of cracks generated after melting, are still lacking. Compared with previous studies, the main novelty of this work is that we studied the effect mechanism of $Cr_3C_2$ content and process parameters on the crack generation of $Fe_3Al/Cr_3C_2$ composites. The process window of the noncracking cladding layer and the preparation method of large-area overlapping coatings without crack generation were also investigated.

## 2. Method

### 2.1. Preparation of Cladding Layer

Carbon structural steel was used as the base material. The cladding powder was a mixture of $Fe_3Al$ and $Cr_3C_2$ powder of particle size 75–125 μm. The $Cr_3C_2$ powder was a commercial pure $Cr_3C_2$ prepared by crushing a large $Cr_3C_2$ block. The chemical composition of $Fe_3Al$ powder is shown in Table 1.

**Table 1.** Chemical composition of $Fe_3Al$ powder (wt.%).

| Al | B | Cr | Fe | Mn | Ni | Zr |
|----|----|----|----|----|----|----|
| 15.23 | 0.11 | 5.78 | 76.82 | 0.052 | 0.54 | 0.26 |

Aluminum blocks and iron powder (CISRI, Beijing, China), applied as raw materials for powder atomization, were melted in a vacuum furnace at 1900~2000 °C for 1 h and then atomized into powder. In this paper, the effects of $Cr_3C_2$ content and process parameters on the cracks of the cladding layer were studied by orthogonal experiment. Four factors, including scanning speed, laser power, powder feed rate, and powder flow, were selected according to the pre-experiment and literature. Five levels were set for each factor. The powder feeding rate of the powder feeder is displayed as an analog value, and the corresponding relationship with the actual powder feed rate is shown in Table 2. The specific levels of each factor are shown in Table 3. There are 25 groups of orthogonal experiments. The specific experimental parameters and the content of $Cr_3C_2$ are shown in Table 4.

**Table 2.** Corresponding table of analog quantity and actual value of powder delivery rate.

| Analog Quantity | Actual Powder Feed Rate (kg/h) |
|-----------------|--------------------------------|
| 6 | 0.322 |
| 8 | 0.429 |
| 10 | 0.537 |
| 11 | 0.590 |
| 12 | 0.644 |
| 13 | 0.698 |
| 14 | 0.751 |
| 16 | 0.966 |

**Table 3.** Factors and levels of orthogonal experiments.

| | Powder Feed Rate (Analog Quantity) | Powder Flow (L/min) | Scanning Speed (m/s) | Laser Power (kW) | $Cr_3C_2$ Content(wt.%) | Blank Column |
|---|---|---|---|---|---|---|
| Level 1 | 10 | 5 | 0.002 | 1.6 | 0 | |
| Level 2 | 11 | 5.5 | 0.0025 | 1.8 | 5 | |
| Level 3 | 12 | 6 | 0.003 | 2 | 15 | |
| Level 4 | 13 | 6.5 | 0.0035 | 2.2 | 25 | |
| Level 5 | 14 | 7 | 0.004 | 2.4 | 35 | |

**Table 4.** Specific parameters of orthogonal experiments.

| Experiment Number | Powder Feed Rate (Analog Quantity) | Powder Flow (L/min) | Scanning Speed (m/s) | Laser Power (kW) | $Cr_3C_2$ Content (wt.%) |
|---|---|---|---|---|---|
| Experiment 1 | 10 | 5 | 0.002 | 1.6 | 0 |
| Experiment 2 | 10 | 5.5 | 0.0025 | 1.8 | 5 |
| Experiment 3 | 10 | 6 | 0.003 | 2 | 15 |
| Experiment 4 | 10 | 6.5 | 0.0035 | 2.2 | 25 |
| Experiment 5 | 10 | 7 | 0.004 | 2.4 | 35 |
| Experiment 6 | 11 | 5 | 0.0025 | 2 | 25 |
| Experiment 7 | 11 | 5.5 | 0.003 | 2.2 | 35 |
| Experiment 8 | 11 | 6 | 0.0035 | 2.4 | 0 |
| Experiment 9 | 11 | 6.5 | 0.004 | 1.6 | 5 |
| Experiment 10 | 11 | 7 | 0.002 | 1.8 | 15 |
| Experiment 11 | 12 | 5 | 0.003 | 2.4 | 5 |
| Experiment 12 | 12 | 5.5 | 0.0035 | 1.6 | 15 |
| Experiment 13 | 12 | 6 | 0.004 | 1.8 | 25 |
| Experiment 14 | 12 | 6.5 | 0.002 | 2 | 35 |
| Experiment 15 | 12 | 7 | 0.0025 | 2.2 | 0 |
| Experiment 16 | 13 | 5 | 0.0035 | 1.8 | 35 |
| Experiment 17 | 13 | 5.5 | 0.004 | 2 | 0 |
| Experiment 18 | 13 | 6 | 0.002 | 2.2 | 5 |
| Experiment 19 | 13 | 6.5 | 0.0025 | 2.4 | 15 |
| Experiment 20 | 13 | 7 | 0.003 | 1.6 | 25 |
| Experiment 21 | 14 | 5 | 0.004 | 2.2 | 15 |
| Experiment 22 | 14 | 5.5 | 0.002 | 2.4 | 25 |
| Experiment 23 | 14 | 6 | 0.0025 | 1.6 | 35 |
| Experiment 24 | 14 | 6.5 | 0.003 | 1.8 | 0 |
| Experiment 25 | 14 | 7 | 0.0035 | 2 | 5 |

### 2.2. Preparation of Cladding Layer

The forming method we used during the research process was laser cladding, and the equipment model selected was the YLS-6000 fiber laser (IPG Photonic, Oxford, MA, USA). The laser's mounting platform was the KUKA mechanical arm, which, together with the powder feeder, formed the forming system, and images and details of the setup are shown in Figure 1. The laser wavelength was 1.07 μm. A Gaussian distribution focusing lens was selected for use on the laser head to generate a rectangular laser spot of size 2 mm × 5 mm. The powder feeding type was coaxial powder feeding, the shielding gas and powder carrying gas were high-purity argon (99.99%), and the flow of shielding gas was 15 L/min. In order to ensure the efficiency and quality of the forming, before cladding, we used 240# sandpaper to polish the surface of the base material and cleaned it with acetone to reduce the brightness of the base material and remove the oil and oxides on the surface.

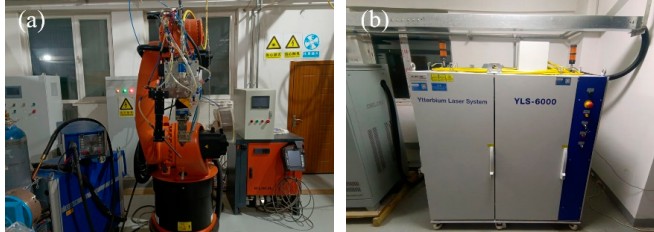

**Figure 1.** Laser cladding system: (**a**) KUKA robotic arm, (**b**) YLS-6000 fiber laser, and (**c**) powder feeder.

*2.3. Microstructure Characterization*

We obtained the cross-section of the cladding layer along the direction perpendicular to the cladding layer by wire cutting. The sample sizes were 10 mm × 10 mm, which is the full thickness of the section. We polished the section with 80#, 240#, 600#, and 1000# sandpaper in turn and then polished the sample with 2.5 um diamond polishing agent. After polishing, the samples were washed and dried. Phase analysis of alloy powders and cladding layers was performed using a Bruker D8 ADVANCE X-ray diffractometer (XRD, BRUKER, Karlsruhe, Germany) at a scanning speed of 2 °/min. In order to observe the microstructure of the cladding layer, a Zeiss high-resolution field emission scanning electron microscope (SEM, ZEISS, Jena, Germany) was used to observe the cross-section of the cladding layer. TEM samples were prepared using an Auriga-focused ion beam Zeiss FIB. Selected observation areas were coated and marked using a vapor deposition system (GIS), and then the samples were sheared to the appropriate thickness using an ion beam. The prepared samples were observed by high-resolution transmission electron microscopy (JEM-2010, JEOL, Tokyo, Japan).

## 3. Result

The macro morphology of the cladding layers prepared by the orthogonal experiment is shown in Figure 2 below. On the whole, the cladding layers are well formed.

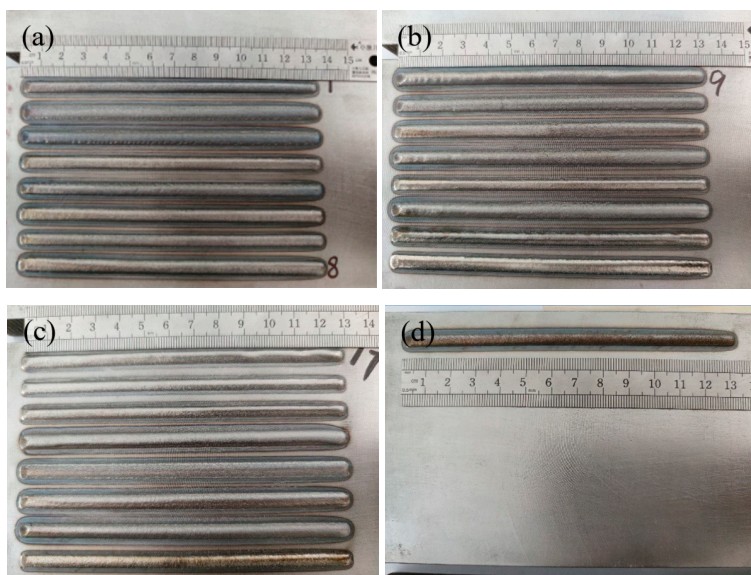

**Figure 2.** Macroscopic morphology of laser cladding. (**a**) The macro morphology of cladding layer of experiment 1–8; (**b**) the macro morphology of cladding layers of experiment 9–16; (**c**) the macro morphology of cladding layer of experiment 17–24; (**d**) the macro morphology of cladding layer of experiment 25.

The crack generation positions of the cladding layers after cladding were observed, as shown in Figure 3. It is found that the main locations of cracks are the heat-affected zone, weld toe, and surface of the cladding layer. It is preliminarily judged that the cracks generated after cladding are cold cracks.

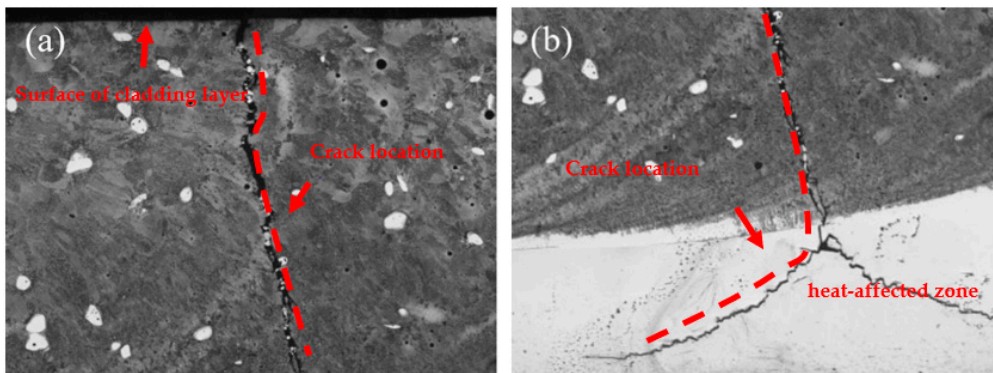

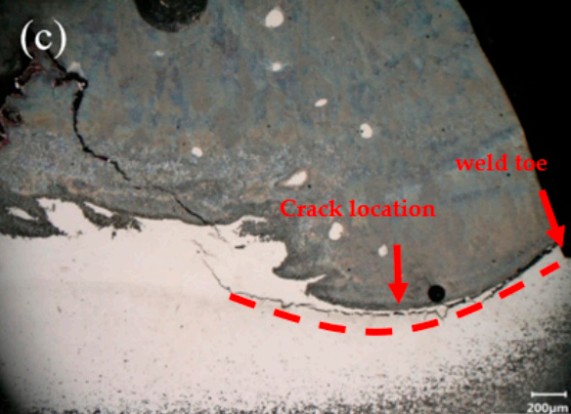

**Figure 3.** Crack generation locations of the cladding layers. (**a**) Cladding surface, (**b**) heat-affected zone, and (**c**) toe welding.

The number of cracks of each cladding layer in the orthogonal experiment was input as the result of orthogonal experiment analysis, and the intuitive analysis data in Table 5 was obtained. Figure 4a–e are curve diagrams of the crack effect of the cladding layer.

**Table 5.** Visual analysis of the number of cracks in cladding layer.

| | Powder Feed Rate | Powder Flow | Scanning Speed | Laser Power | Cr$_3$C$_2$ Content | Blank Column |
|---|---|---|---|---|---|---|
| Mean value 1 | 4.6 | 5.6 | 3 | 8 | 15 | 4 |
| Mean value 2 | 10 | 7.6 | 6.4 | 6.6 | 9.4 | 9.5 |
| Mean value 3 | 4 | 10 | 5 | 7.6 | 0.4 | 7.667 |
| Mean value 4 | 8.6 | 7 | 10.2 | 4.6 | 1 | 9.8 |
| Mean value 5 | 7 | 4 | 9.6 | 7.4 | 8 | 3.6 |
| Range | 6 | 6 | 7.2 | 3.4 | 14.2 | 6.2 |

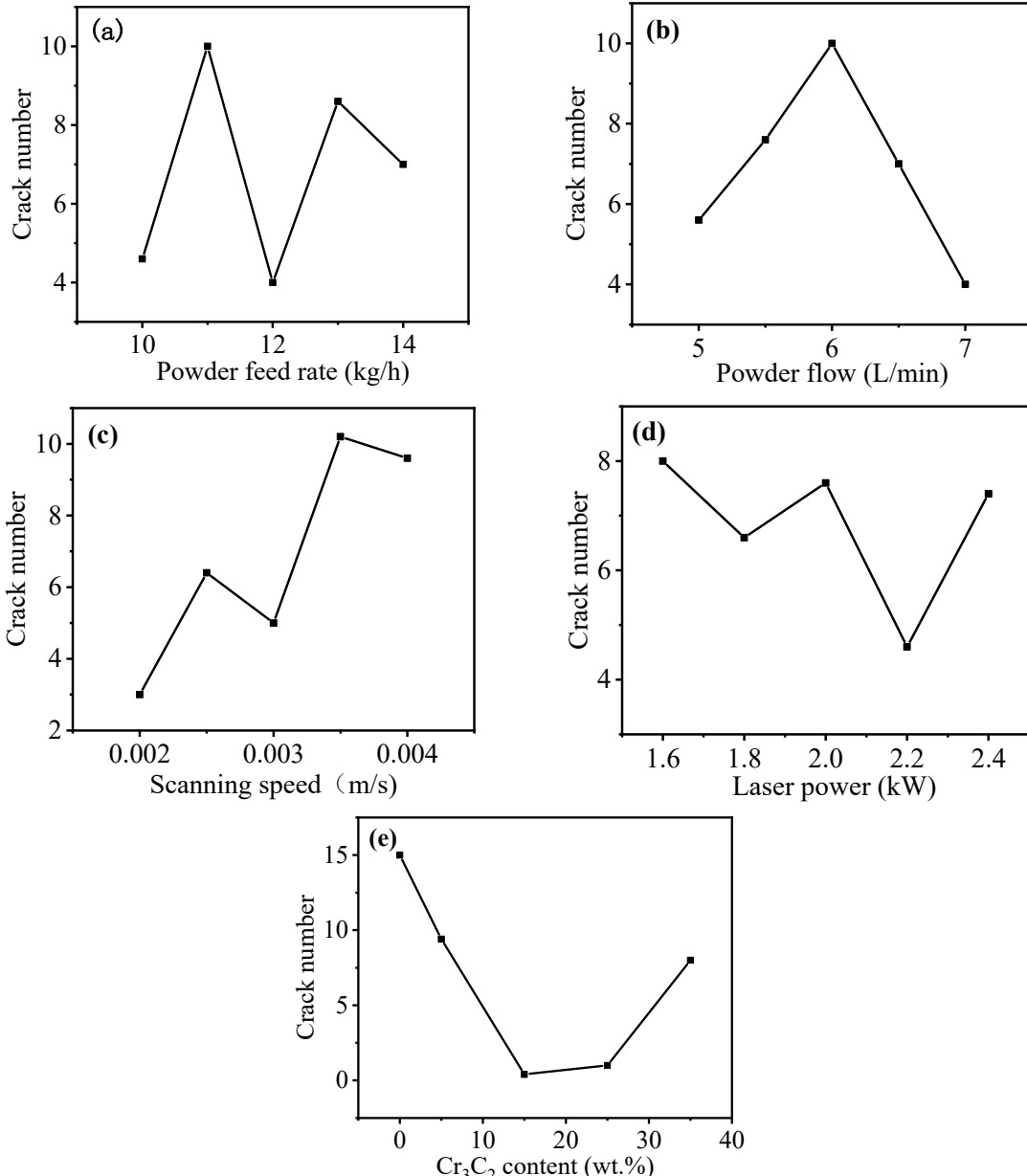

**Figure 4.** Curve diagram of crack effect of cladding layer. (**a**) The effect curve of powder delivery rate on the number of cracks; (**b**) the effect curve of powder delivery flow on the number of cracks; (**c**) the effect curve of scanning rate on the number of cracks; (**d**) the effect curve of laser power on the number of cracks; (**e**) the effect curve of strengthening phase content on the number of cracks.

It can be seen from the intuitive analysis that the most important factor affecting the number of cracks in the cladding layer is the content of $Cr_3C_2$, and the second most important factor is the scanning rate. The effect of laser power on the number of cracks is minimal. The experimental data were then analyzed by effect curve.

As can be seen from the effect curve in Figure 4, as for the main factor affecting cracks, $Cr_3C_2$ content, with the increase in $Cr_3C_2$ content, the number of cracks first significantly decreased and then significantly increased, and the value is the minimum when the content of $Cr_3C_2$ was 15 wt.%. The effect of the scanning rate on the number of cracks basically shows that with the increase in scanning rate, the number of cracks also increases gradually. When the scanning rate is 0.002 m/s, the number of cracks reaches the minimum. The other three factors have relatively little effect on the crack, and the law of change is not obvious with the factors changing.

Therefore, we should first make the microstructure and phase constitution of each cladding layer clear. BSE morphology of cladding layers with different $Cr_3C_2$ contents under the optimal process is observed, and the results obtained are shown in Figure 5. It can be found that the difference in $Cr_3C_2$ content has a great influence on the microstructure of the cladding layer. From Figure 5a, we can find that only a single matrix composition can be observed with no $Cr_3C_2$ added. Figure 5b–d show at 5, 15, and 25 wt.% $Cr_3C_2$ contents, the in situ autogenous strengthening phase is distributed in the matrix phase as a network structure. With the increase in $Cr_3C_2$ content, the reticular structure becomes more and more dense, and the void decreases. Figure 5e,f show at 35 wt.% $Cr_3C_2$ content, the structure of the strengthened phase changes from the original reticular structure to the block-shaped structure.

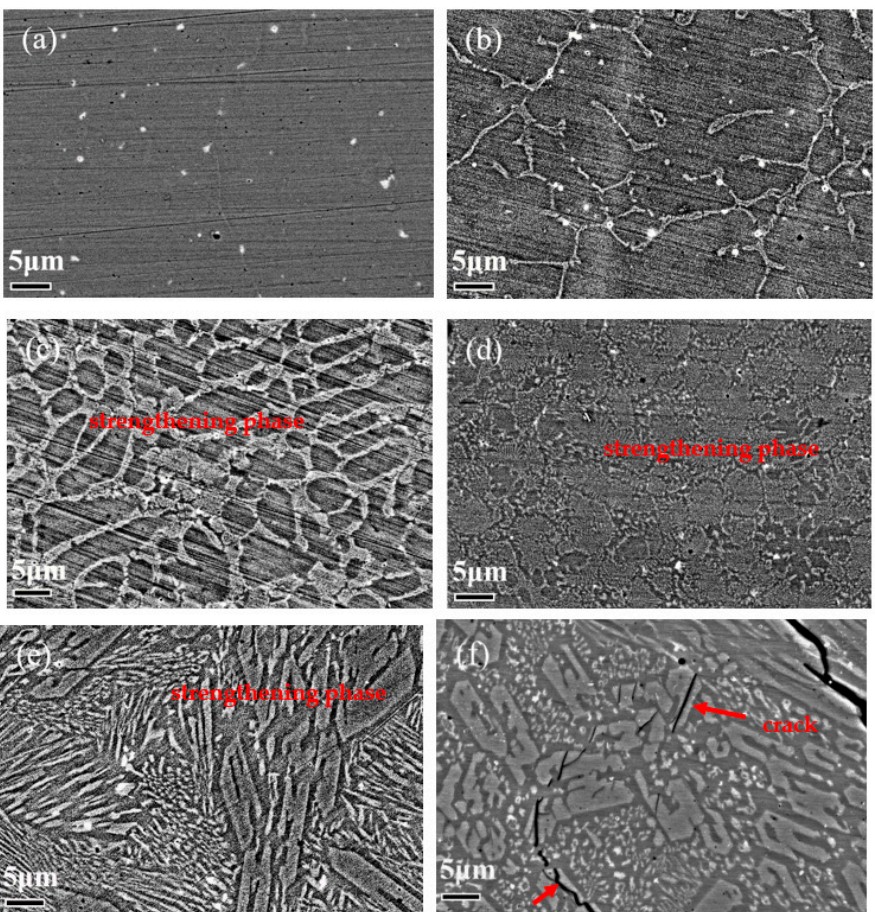

**Figure 5.** BSE morphology of cladding layers prepared using the same process and different $Cr_3C_2$ content: (**a**) 0 wt.%; (**b**) 5 wt.%; (**c**) 15 wt.%; (**d**) 25 wt.%; (**e**) 35 wt.% $Cr_3C_2$; and (**f**) internal microcrack generation location of 35 wt.% $Cr_3C_2$ cladding layer.

In order to figure out the phase in the cladding layer, XRD phase analysis was carried out on the five groups of cladding layers prepared with different $Cr_3C_2$ contents. The wavelength of the X-ray radiation used for the XRD experiments is 1.5418Å, and the results are shown in Figure 6. Figure 6a shows the presence of only one phase of $Fe_3Al$ in the 0 wt.% $Cr_3C_2$ cladding layer. Figure 6b–e show that the 5, 15, 25, and 35 wt.% $Cr_3C_2$ cladding layers contain $Fe_3Al$ (JCPDS 45-1203), $Fe_2AlCr$ (JCPDS 42-1486), and $Cr_7C_3$ (JCPDS 45-1203). Because the peak characteristic of $Fe_3Al$ coincides with that of $Fe_2AlCr$, the existence of an $Fe_2AlCr$ phase could not be determined from the XRD results alone.

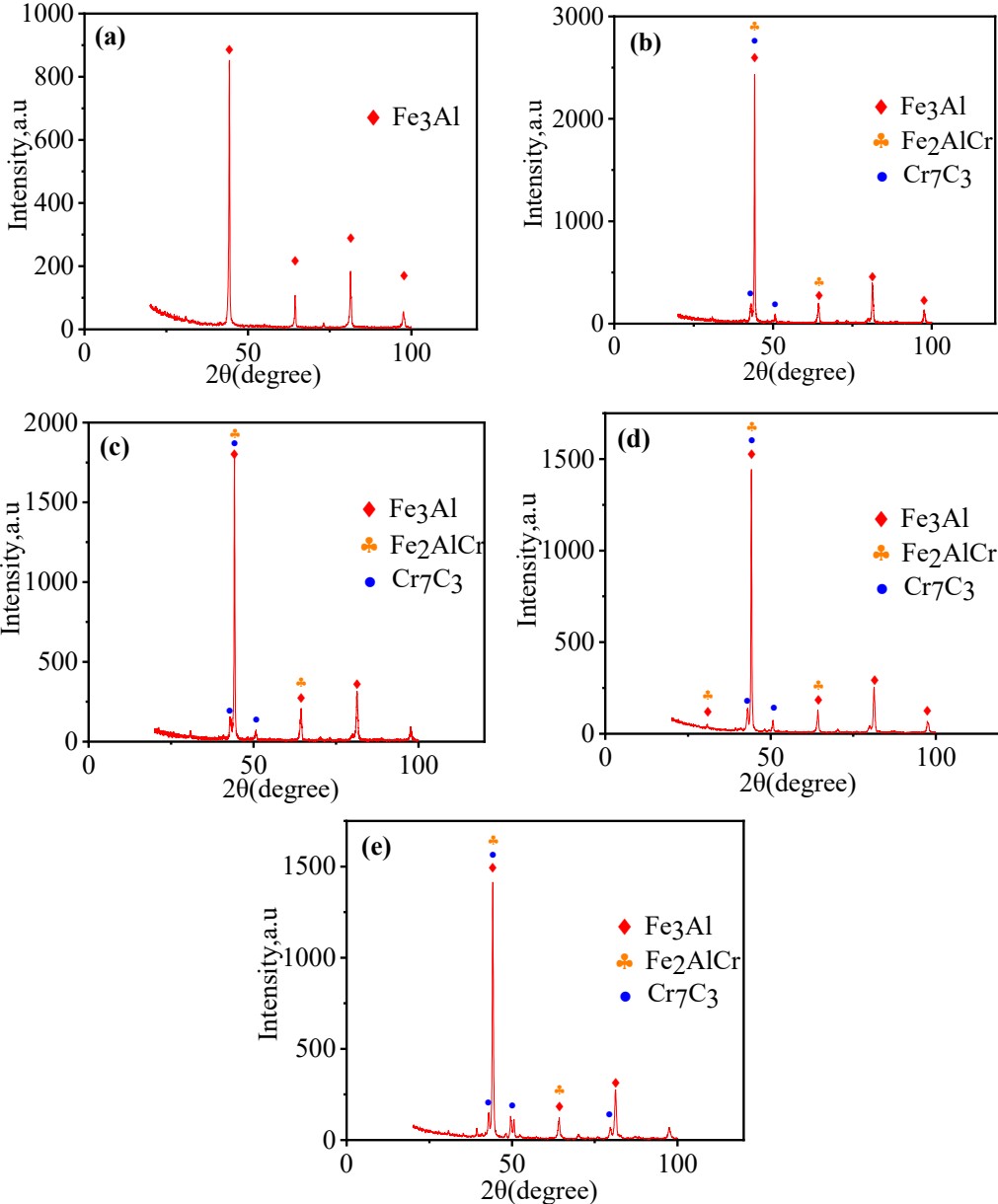

**Figure 6.** XRD patterns of cladding layers with the same process and different Cr$_3$C$_2$ contents: (**a**) 0 wt.%, (**b**) 5 wt.%, (**c**) 15 wt.%, (**d**) 25 wt.%, (**e**) and 35 wt.%.

The dislocation structure of 0 wt.%, 15 wt.%, and 35 wt.% Cr$_3$C$_2$ cladding layers was observed by TEM. Figure 7a shows that the dislocation density of the 0 wt.% cladding layer is very high. Figure 7b,c show the dislocation lines of the 15 wt.% and 35 wt.% Cr$_3$C$_2$ matrix are short and have low density. The smaller dislocation density also means that the two groups of cladding layers have more plastic reserve, which verifies the above experimental results and also reflects that Fe$_2$AlCr makes a greater contribution to the plasticity of cladding layers than Fe$_3$Al.

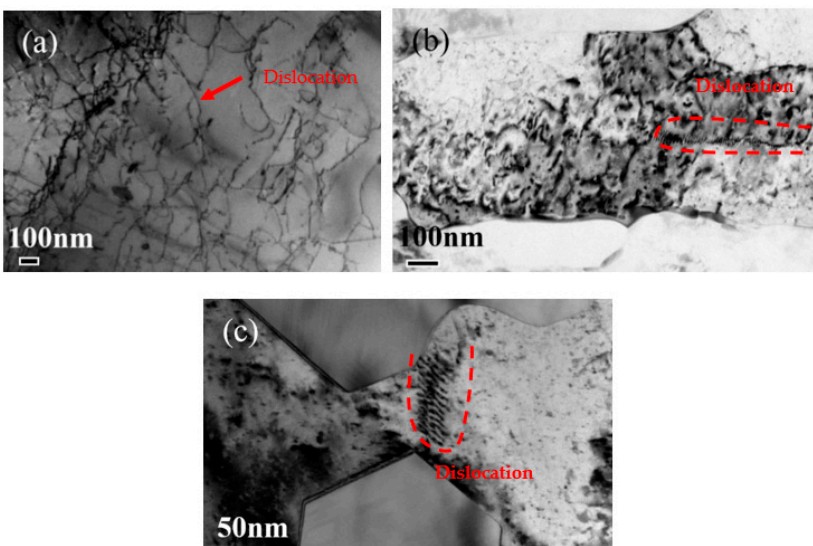

**Figure 7.** Transmission dislocation morphologies of 0 wt.% Cr₃C₂, 15 wt.% Cr₃C₂ and 35 wt.% Cr₃C₂ cladding layers. (**a**) Dislocation topography of 0 wt.% Cr₃C₂ cladding layer; (**b**) dislocation topography of 15 wt.% Cr₃C₂ cladding layer; (**c**) dislocation topography of 35 wt.% Cr₃C₂ cladding layer.

We use back perspective to observe the internal structure of each group of cladding layers. As shown in Figure 8, it can be found that, overall, The internal structure morphology of each group of melt layers is almost the same. The strengthening phase is distributed in the matrix in the form of a network.

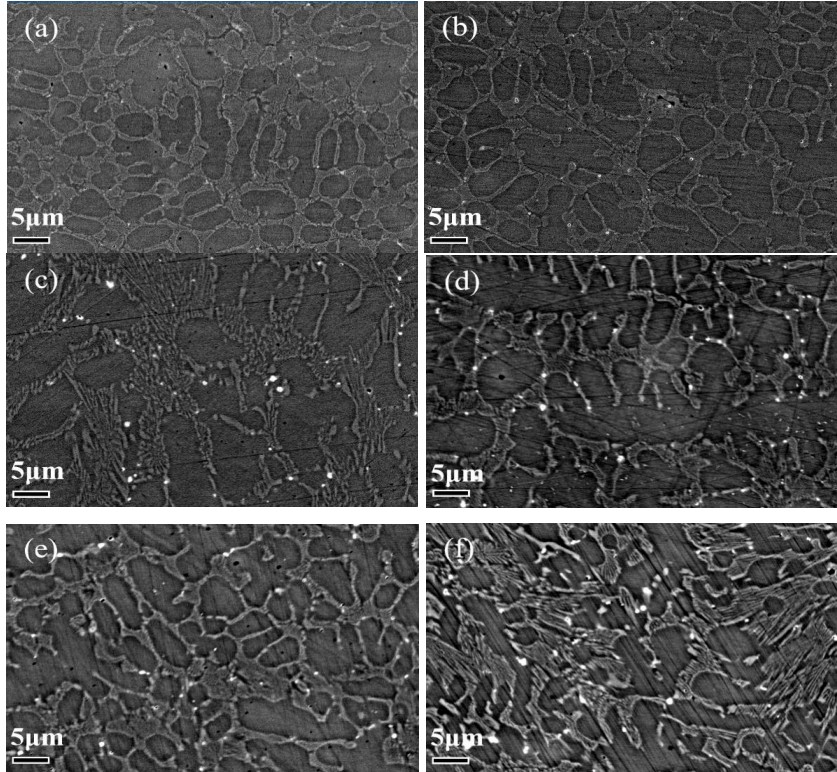

**Figure 8.** BSE morphology of cladding layers prepared using the same Cr₃C₂ content and different process parameters. (**a**) Scanning rate 0.003 m/s (**b**); scanning rate 0.004 m/s; (**c**) laser power 1.8 kW (**d**) laser power 2.6 kW; (**e**) powder feeding rate 8; (**f**) powder feeding rate 16.

XRD phase analysis was carried out on six groups of cladding layers with the same $Cr_3C_2$ content and different processes, and the results are shown in Figure 9. The XRD results show that when the content of $Cr_3C_2$ is 15 wt.%, the phase in the cladding layer does not change significantly with the scanning rate, laser power, and powder feed rate changing independently, and the main phases in the cladding layer are still $Fe_3Al$, $Cr_7C_3$, and $Fe_2AlCr$.

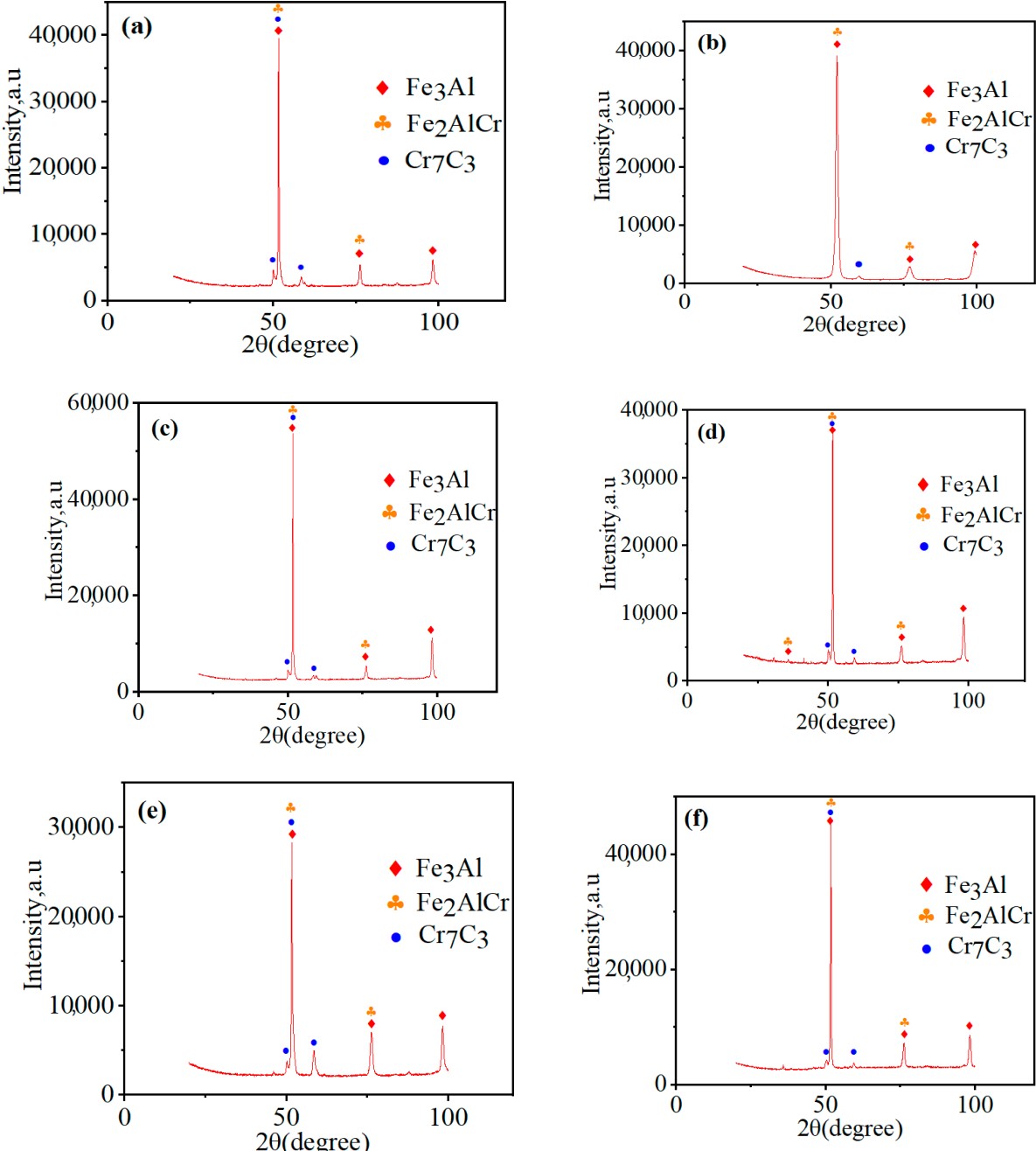

**Figure 9.** XRD patterns of cladding layers with the same $Cr_3C_2$ content and different process parameters: (**a**) scanning rate 0.003 m/s; (**b**) scanning rate 0.004 m/s; (**c**) laser power 1.8kW; (**d**) laser power 2.6 kW; (**e**) powder feeding rate 8; and (**f**) powder feeding rate 16.

EDS energy spectrum measurement was performed on the cladding layer substrate. The results are shown in Table 6.

**Table 6.** Cr content of cladding substrate with the same $Cr_3C_2$ content and different process parameters.

| Laser Power (kW) | Scanning Speed (m/s) | Powder Feed Rate (kg/h) | Crack Number | Cr Content in Matrix (Atomic Percentage) |
|---|---|---|---|---|
| 2.2 | 0.003 | 12 | 0 | 16.16% |
| 2.2 | 0.004 | 12 | 0 | 15.77% |
| 1.8 | 0.002 | 12 | 0 | 17.22% |
| 2.6 | 0.002 | 12 | 2 | 14.94% |
| 2.2 | 0.002 | 16 | 0 | 17.10% |
| 2.2 | 0.002 | 8 | 0 | 15.72% |

By selecting the 15 wt.% and 25 wt.% $Cr_3C_2$ contents, which have a smaller tendency to generate cracks in the previous text, we obtain process windows where there are no obvious macroscopic cracks on the surface of the cladding layer at a scanning rate of 0.002 m/s, as shown in Figure 10.

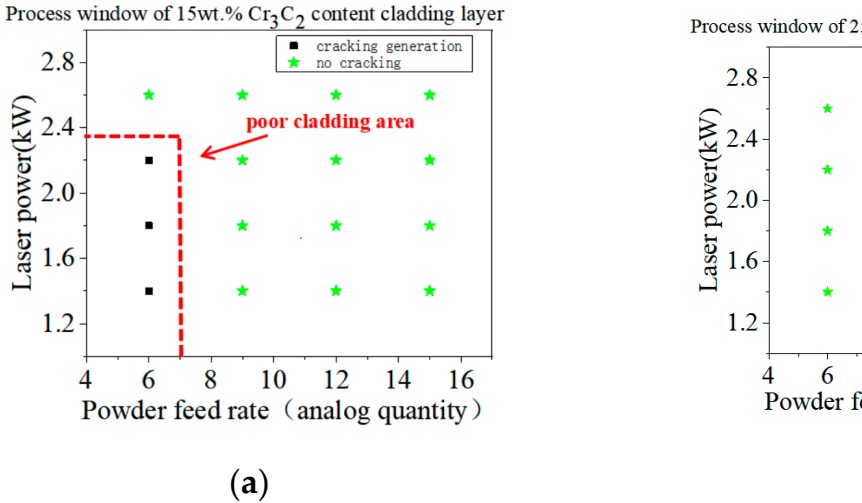
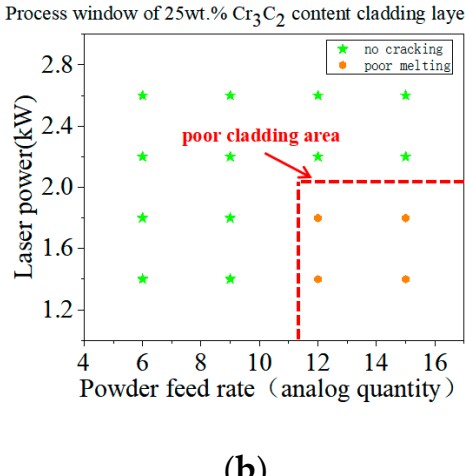

(**a**)                (**b**)

**Figure 10.** Process windows (**a**)15 wt.% $Cr_3C_2$ (**b**) 25 wt.% $Cr_3C_2$

The laser power range for the process window is selected from 1.4 to 2.6 kW. If the laser power is below 1.4 kW, the cladding layer cannot achieve good metallurgical bonding with the substrate. If it is above 2.6 kW, the laser power is too high, which is unfavorable for actual production cost and condition. The selection range of the powder feed rate is 6–15. When it is below six, the cladding layer is too thin, and the cladding efficiency is too low, which does not meet the actual needs of the project. When it is above 15, the cladding layer is too high and will waste material.

In the obtained process window, we selected parameters to prepare a large area of crack-free coating. Process parameters that led to fewer cracks and better forming were determined: $Cr_3C_2$ content of 25 wt.%, laser power of 1.8 kW, laser scanning speed of 0.002 m/s, and powder feed rate of 6. The laser scanning route adopts an s-shaped path, where the starting point of each pass is the endpoint of the previous pass. The final prepared multilayer overlap fusion coating is shown in Figure 11. It can be seen that the surface of the cladding layer prepared under this process and the $Cr_3C_2$ content is bright and flat, with good formation. After penetration testing, there are no macroscopic cracks on the surface.

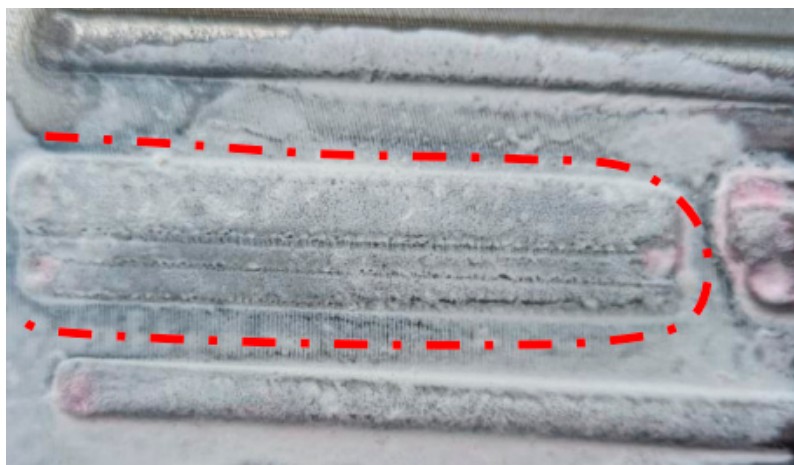

**Figure 11.** Penetration testing result of large area cladding layer.

## 4. Discussion

Cold cracks can be divided into three categories according to their causes. The first category is quenching brittle crack, which usually occurs in hardened composition under stress. When the temperature is low, the crack tends to occur in the composition. Some ultrahigh-strength steels, martensitic stainless steels, and tool steels have a high susceptibility to quenching brittle cracks. The second category is delayed cracking, which does not appear immediately but after cooling for a period of time, and its generation is related to the hardened composition, hydrogen, and stress. The third category of low plastic embrittlement crack is due to the low plasticity of the material itself, resulting in cracks under stress. Such cracks will appear when the temperature is cooled below a certain value, and the weld toe and heat-affected zone are crack-prone zones [8–10].

The evidence supporting that the cracks generated in the cladding layer prepared above are cold cracks was determined as follows: Almost all the cracks are perpendicular to the welding direction, and the edges are sharp. It can be observed that cracks pass through the crystal. Judging from the sound of the crack generation, the crack-forming sound can be heard in the process of cladding and a long time after the end of cladding (one to two days). These are consistent with the characteristics of typical cold cracks.

No matter which kind of cold crack appears, it means that the local position ductility $\delta$min of the cladding layer cannot withstand the strain $\varepsilon$ effect. $\varepsilon$ is related to restraint stress. $\delta$min represents the ability of the composition to withstand inelastic deformation without significant reduction in its withstanding ability before failure, which is related to the factors that cause the material to become brittle during welding cooling, such as quenched composition, hydrogen embrittlement, and the properties of the material itself. Small ductility means that the cladding layer is brittle, and the plastic reserve is insufficient.

Due to the fast-cooling speed of laser cladding, there is a large welding tensile stress inside the cladding layer. With the same process, there is no significant difference between the internal welding tensile stress of the 15 wt.% $Cr_3C_2$ cladding layer and the 0 wt.% $Cr_3C_2$ cladding layer. When the $Cr_3C_2$ content of the cladding layer is 15 wt.%, the carbide phase appears in the $Fe_3Al$ matrix. The interface between different phases is prone to become a stress-concentrated area, leading to a larger cracking tendency [11]. Even so, the cladding layer with 15 wt.% $Cr_3C_2$ still has fewer cracks than the 0 wt.% $Cr_3C_2$ cladding layer, indicating that the increase in plastic reserve of the cladding layer leads to the suppression of cracks. Next, the reasons for the increase in cladding plasticity are analyzed. By comparing the microstructure of the 0 wt.% and 15 wt.% $Cr_3C_2$ cladding layers, it is found that the possible reasons for the reduction in cracks in the 15 wt.% $Cr_3C_2$ cladding layer are as follows: (1) There is carbide strengthening phase in the cladding layer, which increases the overall plasticity and reduces the crack generation. (2) The addition of 15 wt.% $Cr_3C_2$ strengthens the plasticity of the matrix material $Fe_3Al$ during the preparation and

reduces the crack. If it is the strengthening phase or $Cr_3C_2$ phase that causes the increase in plasticity, it can be seen from the crack location of the 35 wt.% $Cr_3C_2$ cladding layer in Figure 5f that the strengthening phase itself is brittle and cannot improve the whole material plasticity. Therefore, it can be deduced that Cr or C elements enter into the $Fe_3Al$ matrix, which increases the plasticity of the matrix. Further analysis shows that among Cr and C elements, C does not contribute much to the plasticity of the cladding layer. On the contrary, it is easy to form a more brittle carbide phase, resulting in increased brittleness of the cladding layer. Therefore, Cr content is the key to improving the plasticity of $Fe_3Al/Cr_3C_2$ composite material.

The main elements in the matrix of the cladding layer are Fe, Al, Cr, and C. With the increase in the content of $Cr_3C_2$, the content of Cr in the matrix showed a tendency of nearly linear increase. The percentage of Cr atom in the $Fe_3Al$ matrix is 16.5% with an optimum $Cr_3C_2$ content of 15 wt.%. Therefore, it can be considered that when the concentration of Cr in the $Fe_3Al$ matrix is near this value, the overall plasticity of the cladding layer is the best, which is conducive to resisting the generation of cracks. When Cr content is far from 16.5%, whether it is much lower than 16.5% or much higher than 16.5%, the plasticity of the cladding layer will be relatively reduced.

In the cladding layer group of different process parameters, the maximum value of the powder feed rate is twice the minimum value (16 and 8), the maximum value of the scanning rate is twice the minimum value (0.002 m/s and 0.004 m/s), and the maximum value of the laser power is 1.44 times of the minimum value. Even though the difference in these three factors is so obvious, except for the group with the laser power of 2.6 kW, the change in Cr content in the cladding layer matrix is still small, and the cladding layer does not generate obvious macroscopic cracks. This indicates that when the $Cr_3C_2$ content is fixed, the change in process parameters has little effect on the Cr content in the matrix. That is, under the previously determined optimal $Cr_3C_2$ conditions, the crack generation tendency of the cladding layer is low and is not sensitive to changes in process parameters. It also verifies that in the orthogonal experiment, the influence of process parameters is relatively low compared to the $Cr_3C_2$ content, and the impact on cracks is relatively small.

According to the results of XRD and TEM phase composition analysis in the cladding layer above, except for the 0 wt.% $Cr_3C_2$ cladding layer, its composition is only $Fe_3Al$, while the composition of other cladding layers is shown as $Fe_3Al$ and $Cr_7C_3$, and may contain $Fe_2AlCr$. $Fe_3Al$ has intrinsic brittleness [12]; this can be verified from the fact that the 0 wt.% $Cr_3C_2$ cladding layer generates many cracks. The cladding layer prepared by pure $Fe_3Al$ is brittle, which makes it very easy to generate low plastic brittle cracks. The number of cracks of the 15 wt.% and 25 wt.% $Cr_3C_2$ cladding layer decreased significantly, which indicates that the plastic reserve increased. This is caused by the newly emerged $Cr_7C_3$ or $Fe_2AlCr$ phases. $Cr_7C_3$ is carbide, which has high hardness and brittleness, as mentioned above. The carbide phase is prone to become the crack source in the cladding layer, which will not increase but decrease the plastic reserve. Through analysis, we can know that the $Fe_2AlCr$ phase must exist and is the reason for the plastic increase in the cladding layer. In the 0 wt.%, 5 wt.%, and 15 wt.% $Cr_3C_2$ cladding layer, Cr content in the matrix increases gradually with the increase in $Cr_3C_2$, and more $Fe_3Al$ is converted to $Fe_2AlCr$ in the $Fe_3Al$ matrix, and the content of carbide is still at a low level. Therefore, the cladding layer is mainly characterized by $Fe_2AlCr$, increasing the plasticity and reducing the crack generation. When the $Cr_3C_2$ content is 15 wt.%, 25 wt.%, and 35 wt.%, the content of $Fe_2AlCr$ is also increased with the increase in $Cr_3C_2$, but due to the introduction of too much $Cr_3C_2$, too much carbide phase is formed in the cladding layer. Although the plasticity of the matrix material increases, the bulk carbide phase occupies most of the volume, so overall, the brittleness of the cladding layer still increases. The brittleness of carbide is the main factor affecting the properties of the cladding layer, resulting in an increase in the number of cracks. At the same time, the content of Cr in the $Fe_3Al$ matrix is not the more the better. Studies have shown that [13] the addition of too much Cr will also increase the stress concentration at grain boundaries to increase the crack generation.

In the six groups with the same $Cr_3C_2$ content and different processes, cracks only appear in the cladding layer with a laser power of 2.6 kW. Combined with the previous analysis, it can be found that the excessive power leads to the larger dilution rate of the cladding layer, which leads to the reduction in Cr content in the matrix material and thus reduces the content of $Fe_2AlCr$. The plastic of the cladding layer is reduced, and the cladding layer generates cracks.

The causes of crack generation in the cladding layer with different $Cr_3C_2$ content or different processes discussed above are essentially about whether a proportion of $Fe_2AlCr$, $Fe_3Al$, and $Cr_7C_3$ content matches. If the content of $Cr_3C_2$ is too low or the dilution rate is too large, the content of $Fe_2AlCr$ in the matrix will be low. Too high $Cr_3C_2$ content and too low dilution rate will lead to high $Cr_7C_3$ strengthening phase content, which will induce the generation of cracks.

During the preparation process of the $Fe_3Al/Cr_3C_2$ cladding layer, part of the Fe atom in $Fe_3Al$ is replaced by the Cr atom, and the original Fe–Al bond is transformed into the Cr–Al bond of $Fe_2AlCr$. The dissociation energy of the atomic pair between dissimilar atoms is calculated [14]: The value of the Cr–Al pair is 0.6960 eV, less than that of the Fe–Al pair 0.7457 eV. This indicates that the interaction between Cr–Al atoms is weak; that is, the addition of Cr reduces the binding energy between the next-neighbor atoms of the $Fe_3Al$ intermetallic compound with $D0_3$ structure, thus reducing the reverse phase domain boundary energy of the alloy and making cross slip more likely to occur, thus improving the plasticity of the cladding layer [14].

From the 15 wt.% $Cr_3C_2$ content process window, it can be seen that cracks will only appear in the cladding layer at a lower powder feed rate of 6. At this ultralow powder feed rate, laser energy not only melts all the cladding powder but also most of the remaining energy melts the base material into a molten pool, resulting in a big dilution rate of the final prepared cladding layer and a lot of base material components entering the cladding layer. With other powder feed rates, no cracks were observed in the power laser range of 1.4 to 2.6 kW. When the $Cr_3C_2$ content is 25 wt.%, even at a small powder feed rate, the cladding layer can still maintain no cracks. At this point, although the dilution rate of the cladding layer is relatively high, the increase in $Cr_3C_2$ content ensures the Cr content in the matrix, which is the $Fe_2AlCr$ content, so that the cladding layer remains with no cracks.

Through the adjustment of composition and process, this material can realize the preparation of large-area crack-free coatings, which provides the possibility for industrial application. Due to the excellent wear resistance and corrosion resistance of the $Fe_3Al/Cr_3C_2$ composite material, it can be popularized and applied to many equipment in the coal mine field in the future, such as scraper machines, coal shearers, drill pipes, etc.

## 5. Conclusions

In this paper, the effects of $Cr_3C_2$ content and process parameters on the cracks of the cladding layer were studied by orthogonal experiment. The optimal process parameters, which can obtain the least cracks, are predicted and verified by an orthogonal test. The cladding layers with different $Cr_3C_2$ content and processes were characterized by TEM. The phase composition of each cladding layer is analyzed. The causes of crack formation and improvement in the cladding layer are analyzed.

1.  When $Cr_3C_2$ is not added, the cladding layer contains a single and uniform composition. With the increase in $Cr_3C_2$ content, the strengthening phase begins to appear and changes from a sparse network to a dense reticular structure and, finally, to a large block structure. The main phases are $Fe_3Al$, $Fe_2AlCr$, and $Cr_7C_3$. At a specific $Cr_3C_2$ content, changing the process parameters has little effect on the microstructure and phase composition of the cladding layer.
2.  The influence of $Cr_3C_2$ content and process parameters on the crack number of the cladding layer was studied. Through an orthogonal test, it is found that the content of $Cr_3C_2$ is the primary factor affecting the crack, and the process parameters have little influence on the crack. When the content of $Cr_3C_2$ increased, the number of cracks

showed a law of significant decrease at first and then significantly increased, and the number of cracks reached the lowest value at 15 wt.%.

3.  The influence of $Cr_3C_2$ content and cladding process parameters on the crack generation was analyzed. Cracks mainly occur in the heat-affected zone, weld toe, and cladding surface, and the crack edge is sharp and passes through the crystal, which is a cold crack. The crack is mainly caused by the mismatch between the plastic phase of $Fe_2AlCr$, the brittle phase of $Fe_3Al$, and $Cr_7C_3$ in the cladding layer. Moreover, with the increase in $Cr_3C_2$ content in cladding powder, $Fe_2AlCr$ and $Cr_7C_3$ in the cladding layer increased, while $Fe_3Al$ decreased. When the content of $Cr_7C_3$ is low, the cladding layer mainly reflects the plastic increase brought by $Fe_2AlCr$; when the content of $Cr_7C_3$ is too much, the property of the cladding layer is reflected as the brittleness of $Cr_7C_3$. The process parameters mainly affect the dilution rate of the cladding layer and thus affect the content of $Fe_2AlCr$.

**Author Contributions:** Conceptualization, L.Z.; Methodology, Y.F.; Software, J.Z.; Formal analysis, Y.F., J.Z. and Z.L.; Writing – original draft, Y.F.; Writing – review & editing, S.L. and L.Z. All authors have read and agreed to the published version of the manuscript.

**Funding:** This work is supported by the Key Technology Innovation Project of China Coal Science and Technology Group, grant number 2020-2-TD-ZD004.

**Institutional Review Board Statement:** Not applicable.

**Informed Consent Statement:** Not applicable.

**Data Availability Statement:** All data that support the findings of this study are included within the article.

**Conflicts of Interest:** The authors declare no conflict of interest.

## Nomenclature

| | |
|---|---|
| SEM | Scanning Electron Microscopy |
| TEM | Transmission Electron Microscopy |
| FIB | Focused Ion Beam |
| XRD | X-ray Diffraction |
| BSE | Backscattered Electron |

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
