# Peer review of "Influential Mechanism of Cr3C2 Content and Process Parameters on Crack Generation of Fe3Al/Cr3C2 Composites"

_coatings, doi:10.3390/coatings13091636_

Round 1

Reviewer 1 Report

The author has presented a very interesting work for the research community, but it is required to incorporate suggestions/comments in their manuscript for easy understanding by researchers. These are as follows.

1. Title should be Influential mechanism of Cr3C2 content and process parameters on

crack generation of Fe3Al/Cr3C2 composites. 

2. The content of  Cr3C2 can be added in the abstract. Further, it should be more quantitative in terms of the results obtained.

3. The author only adds the reference up to 2021. It seems no work has been carried out in 2 years, but it is very doubtful. The author must add a few recent research from 2023 and 2022. Then elaborate on the objective of the work.

4. How the author has control the distortion in cladding?

5. What is the uncertainty of obtained data from Fig 4? How many times test is repeated?

6. The author should add the work's future scope with proper application. 

Author Response

Comment1. Title should be Influential mechanism of Cr3C2 content and process parameters on crack generation of Fe3Al/Cr3C2 composites. 

Response1: The tittle has been modified.

Comment2. The content of  Cr3C2 can be added in the abstract. Further, it should be more quantitative in terms of the results obtained.

Response2:The specific value of the optimal Cr3C2 content to reduce crack generation has been added to the abstract.

Comment3. The author only adds the reference up to 2021. It seems no work has been carried out in 2 years, but it is very doubtful. The author must add a few recent research from 2023 and 2022. Then elaborate on the objective of the work.

Response3:Relevant research on Fe3Al/Cr3C2 composite materials has not been found in the past year. This material is relatively new. I published my first paper about this material in October 2022.

Comment4. How the author has control the distortion in cladding?

Response4:Selecting a more appropriate process parameter range through pre-experimentation can avoid the occurrence of deformation in most cases.

Comment5. What is the uncertainty of obtained data from Fig 4? How many times test is repeated?

Response5:Orthogonal experiment does not need to repeat the experiment and consider the problem of uncertainty. It itself includes multiple groups of experiments. Through the intuitive analysis of the results of multiple groups of experiments, more accurate analysis results can be obtained.

Comment6. The author should add the work's future scope with proper application. 

Response6: This part has been added on line 327-331.

Reviewer 2 Report

The manuscript discusses the formation of composite Fe3Al/Cr3C2 layers fabricated by laser cladding, as well as the influence of the process parameters and amount of Cr3C2 on crack generation. The results showed that the content of Cr3C2 has a major influence on crack formation. Overall, the paper is interesting. I have the following remarks:

- Table 2 is not cited in the text. The data presented there should be described in more detail. 

- The inscriptions available in Figures 3 and 5 are not readable. 

- The wavelength of the X-ray radiation used for the XRD experiments should be presented. Also, the numbers of the ICDD cards used for the XRD results interpretation should be provided.

- The existence of the Fe2AlCr phase has to be confirmed by another characterization method since both detected peaks are common with the Fe3Al phase. Even so, the first one is common for the three phases. 

Author Response

Comment1:Table 2 is not cited in the text. The data presented there should be described in more detail. 

Response1:Table 2 is  cited in the text. The data is described on line 67-69.

Comment2:The inscriptions available in Figures 3 and 5 are not readable. 

Response2: This part has been modified.

Comment3:The wavelength of the X-ray radiation used for the XRD experiments should be presented. Also, the numbers of the ICDD cards used for the XRD results interpretation should be provided.

Response3: This part has been added on line 154-158.

Comment4:The existence of the Fe2AlCr phase has to be confirmed by another characterization method since both detected peaks are common with the Fe3Al phase. Even so, the first one is common for the three phases. 

Response4:We took photos of the selected electron diffraction patterns, but still found that Fe3Al and Fe2AlCr could not be distinguished, because the Fe atoms were too similar to the Cr atoms.

Round 2

Reviewer 1 Report

No Comments